# Nitrogen-Doped Titanium Dioxide as a Hole Transport Layer for High-Efficiency Formamidinium Perovskite Solar Cells

**DOI:** 10.3390/molecules27227927

**Published:** 2022-11-16

**Authors:** Nitin Ralph Pochont, Yendaluru Raja Sekhar, Kuraganti Vasu, Rajan Jose

**Affiliations:** 1School of Mechanical Engineering, Vellore Institute of Technology, Vellore 632014, India; 2Centre for Disaster Mitigation and Management, Vellore Institute of Technology, Vellore 632014, India; 3Department of Physics, School of Advanced Sciences, Vellore Institute of Technology, Vellore 632014, India; 4Center for Advanced Intelligent Materials, Universiti Malaysia Pahang, Kuantan 26300, Malaysia; 5Faculty of Industrial Sciences & Technology, Universiti Malaysia Pahang, Kuantan 26300, Malaysia

**Keywords:** perovskite solar cell, n-i-p structure, nitrogen-doped titanium dioxide, hole transport layer, formamidinium recipe, SCAPS simulation

## Abstract

Perovskite solar cells (PSCs) offer advantages over widely deployed silicon solar cells in terms of ease of fabrication; however, the device is still under rigorous materials optimization for cell performance, stability, and cost. In this work, we explore a version of a PSC by replacing the polymeric hole transport layer (HTL) such as Spiro-OMeTAD, P3HT, and PEDOT: PSS with a more air-stable metal oxide, viz., nitrogen-doped titanium dioxide (TiO_2_:N). Numerical simulations on formamidinium (FA)-based PSCs in the FTO/TiO_2_/FAPbI_3_/Ag configuration have been carried out to depict the behaviour of the HTL as well as the effect of absorber layer thickness (∆t) on photovoltaic parameters. The results show that the cell output increases when the HTL bandgap increases from 2.5 to 3.0 eV. By optimizing the absorber layer thickness and the gradient in defect density (Nt), the device structure considered here can deliver a maximum power conversion efficiency of ~21.38% for a lower HTL bandgap (~2.5 eV) and ~26.99% for a higher HTL bandgap of ~3.0 eV. The results are validated by reproducing the performance of PSCs employing commonly used polymeric HTLs, viz. Spiro-OMeTAD, P3HT, and PEDOT: PSS as well as high power conversion efficiency in the highly crystalline perovskite layer. Therefore, the present study provides high-performing, cost-effective PSCs using TiO_2_:N.

## 1. Introduction

The demand for energy and crises of fossil fuels in today’s world has led to the interest in alternative energy sources [1]. Solar energy offers a clean mode of energy generation with a broad domain of various user applications. To aspire to the energy sustenance goals for 2030, researchers worldwide have explored energy generation sources using renewables. Solar photovoltaics is deemed a well-known renewable source with the potential to meet the demands and sustain its existence [2]. The evolution of solar cells using sophisticated material has led to a technological revolution for developing energy generation solutions in diverse applications such as buildings, aircraft, and satellites. With advanced semiconductor physics and the accessibility to diverse materials, modern photovoltaic solar cells have raised the barrier by attaining more efficient outputs with longer life sustainability. Photovoltaic technology is predominant with the crystalline silicon solar cells that render the opportunity for efficient energy generation on the commercial platform [3,4]. The other classification of solar cells, i.e., thin films such as organic cells and perovskite cells, can be developed cost-effectively with technological advancement using low-cost, flexible substrates to ascertain efficient outputs [5]. Perovskite solar cells (PSCs) are categorized as third-generation thin-film solar cells that counter the Shockley Queisser Limit (SQL) under laboratory conditions [6]. In addition, PSCs acquire intrinsic properties such as tunable bandgap, longer diffusion length, and suitable carrier transport mechanism, which are processable at lower fabrication costs [7]. Hence, modern research confers PSCs as desired photovoltaic devices that are significant in achieving energy sustenance [8]. Researchers have recently developed unique recipes that achieved a power conversion efficiency (PCE) of 25.2%, which was the apex reported in 2020 [9]. Further findings show that PSCs reached an efficiency of 26.1% in 2021, and a multi-junction PSC with a Si-tandem structure attained an efficiency of 29.8% [10]. Henceforth, the recipe or chemical composition of the absorber and supporting layers possess crucial roles that foster the cell output. Intuitively, the stability and upscaling properties of the cell are distinct concepts that are critically considered while fabricating the cell [11].

Laboratory-scale development of the ABX_3_-driven metal halide organic and inorganic perovskite cell structure began in 2009 [3]. Methylammonium lead halide (MAPbX_3_) and formamidinium lead halide (FAPbX_3_) are most endorsed in formulating the cell recipe in a planar or mesoporous structure. The metals (B) and anions (X) are engineered with different elements in view to enhance stability and improve cell efficiency [12]. One major hurdle is the presence of lead (Pb), which increases the toxicity of the cell. So, several studies reported lead being replaced with group IVA and VA elements such as Tin (Sn^2+^), Germanium (Ge^2+^), and Antimony (Sb^2+^) [7,13,14]. On the other hand, organic elements (MA and FA) have depicted an intrinsically unstable behavior, which led to the presage of inorganic elements such as cesium (Cs), rubidium (Rb), and potassium (K), respectively [15,16]. Most of the recipes in the literature for PSC development reported the anionic halogen element as Cl^−^, Br^−^, and I^−^ [17].

The perovskite absorber layer is held between the electron transport layer (ETL) and hole transport layer (HTL) in most of the PSC configurations that act as profound charge transport mediums [18]. According to semiconductor physics, ETL and HTL provide carrier separation paths responsible for avoiding recombination, boosting charge transportation, preventing cell degradation, and efficiently transmitting light. The arrangement of ETL and HTL in a perovskite solar cell is either the n-i-p or p-i-n configuration [19]. These transport layers also act as blocking layers to their opposite counterparts, which help in improving cell stability and life. HTLs are chosen based on the organic and inorganic material band structure and band edge position. Organic HTLs are advantageous over inorganic HTLs in biodegradability and layer processing [20]. However, the drawbacks such as instability, high cost, and multi-step synthesis have led to the adoption of inorganic HTLs. In addition, inorganic HTLs promote appealing features such as hole mobility, chemical stability, and low cost [21]. Typical organic HTLs such as Spiro-OMeTAD, PEDOT: PSS, P3HT, and PTTA; and inorganic HTLs such as CuO_x_, CuSCN, CuI, NiO_x_, MoS_2_, and WS_2_ are widely employed in the PSC structure [22].

The selection of the suitable ETL during cell fabrication is based on efficient electron extraction ability and stability. Due to their surface and electrical properties, the most common ETLs preferred for PSC development are TiO_2_, ZnO, and SnO_2_ [23]. In the recent past, extensive research has been carried out in modulating different ETLs suitable for PSCs to envisage better energy yield. For example, Bendib et al. [24] numerically simulated a P3HT/MAPbI_3_ perovskite structure with ZnSe and ZnS as the ETLs. In contrast, Hima and Lakhdar [25] developed a CH_3_NH_3_GeI_3_ cell structure with C60 as the ETL that yields 23.58% PCE. Bhavsar and Lapsiwala [26] also performed numerical simulations on a Cu_2_O/MAPbI_3_ cell with different ETLs such as PCBM, CdZnS, WS_2_, IGZO, and CdS that yielded a PCE lower than TiO_2_, ZnO, and SnO_2_. Research reveals that ETLs derived from titanium and tin oxide have resulted in stable operation and consistent efficiencies [27,28].

Titanium dioxide (TiO_2_) emerges as a practical and prevalent photocatalyst with chemical stability, nontoxicity, and low cost [29]. Furthermore, cationic or anionic doping modifies the bandgap, optical, and electrical properties of TiO_2_ [30]. With tunable bandgap and Fermi-level shift, TiO_2_ doped with various elements has been demonstrated to be a good ETL with improved efficiency and cell parameters such as open-circuit voltage (V_oc_) [31]. Consequently, the limitation of TiO_2_-based ETL is hindered by its poor absorption of visible light in the solar spectrum. Hence, the development of nitrogen-doped TiO_2_ (ETL) with reduced bandgap possesses photocatalytic properties due to enhanced visible light absorption and reduced recombination rate [32]. Compared to other anion dopants, such as sulphur and phosphorus, nitrogen is a suitable doping element in TiO_2_ that forms a metastable center, reduced atom size, and low ionization energy [33].

Interestingly, a breakthrough was recently reported by Panepinto et al. [34] in devising a nitrogen-doped TiO_2_ (TiO_2_:N) layer as an HTL for application in dye-sensitized solar cells. The p-type TiO_2_:N layer was deposited through co-reactive magnetron sputtering by tuning the O_2_ and N_2_ reactive gases mixture. The bandgap of the HTL is tuneable, and the value depends on the dopant concentration (%) of nitrogen. However, the usage of TiO_2_:N as a low-cost HTL in perovskite solar cell structures has not been explored. With enhanced photocatalytic properties, the applicability of TiO_2_:N as an HTL is suitable for semi-transparent and transparent PSCs as the bandgap is tunable.

We present a novel PSC configuration with TiO_2_:N as the HTL and undoped TiO_2_ as the ETL in a planar formamidinium lead halide recipe by SCAPS numerical simulation. The PSC recipe that yields better performance in terms of PCE corresponding to the variance in absorber layer thickness (∆t) and defect density (∆N_t_) is put forward. In future research, this concept would foster the feasibility of developing stable and semi-transparent PSCs with a low-cost HTL. The influence of physical changes on device performance concerning doping gradient, doping composition, and interface defect density is reported in this research.

## 2. Methodology

### 2.1. SCAPS Simulation

The solar cells capacitance simulator (SCAPS 3.10) one-dimensional software is a modern computational tool developed to simulate solar cell physics numerically. SCAPS provides a theoretical understanding of solar cell behaviour that helps compare results with experimental analysis [35]. SCAPS’s built-in program is designed to numerically solve semiconductor equations in 1-D steady-state conditions [7]. Global researchers recognized SCAPS as a suitable analytical tool to determine I-V characteristics, fill factor, band diagrams, quantum efficiencies, spectral responses, current-voltage density, PCE, and recombination profile within the charge transport layers [26,36,37]. The performance of the fabricated solar cell is persisted based on the semiconductor Equations (1)–(3) [11];
(1)$$Electron\;continuity\;eqn.,\;\;\frac{dn_{p}}{dt}=G_{n}-\frac{n_{p}-n_{po}}{\tau_{n}}+n_{p\;}\mu_{n}\frac{d\xi }{dx}+\mu_{n}\xi \frac{dn_{p}}{dx}+D_{n}\frac{d^{2}n_{p}}{dx^{2}}$$
(2)$$Hole\;continuity\;eqn.,\;\;\frac{dp_{n}}{dt}=G_{p}-\frac{p_{n}-p_{no}}{\tau_{p}}-p_{n\;}\mu_{p}\frac{d\xi }{dx}-\mu_{p}\xi \frac{dp_{n}}{dx}+D_{p}\frac{d^{2}p_{n}}{dx^{2}}$$
(3)$$Poissons\;eqn.,\;\frac{d}{dx}\left(-\varepsilon \left(x\right)\frac{d\psi }{dx}\right)=q\;\left[p\left(x\right)-n\left(x\right)+N_{d}^{+}\left(x\right)-N_{a}^{-}\left(x\right)+p_{t}\left(x\right)-n_{t}\left(x\right)\right]$$

### 2.2. TiO_2_:N as a p-Type HTL

As discussed in the introduction, the most commonly adopted HTLs are organic or inorganic materials. TiO_2_ is a widely used ETL in PSC structures designed so far. As an alternative to expensive HTLs, this research examines the applicability and feasibility of implementing TiO_2_:N as an HTL. However, the validation that confers nitrogen-doped TiO_2_ as a suitable HTL for thin-film solar cells is a point to prove. Several studies revealed that N-doped TiO_2_ exhibits stable p-type conductivity. Vasu et al. [38] employed the atomic layer deposition technique to develop a p-type epitaxial N-doped TiO_2_ thin film. The results depict a reduced optical bandgap and better hole concentration and mobility. Towards analysing the cation vacancies in TiO_2_, Lee et al. [39] reported that n-type TiO_2_ and p-type TiO_2_ exhibit similar morphology, surface area, and crystal structure. Comparatively, p-type TiO_2_ has better stability and performance rate. Outwardly, Vasilopoulou et al. [40] stated that p-type nitrogen doping enhances the photocatalytic efficiency of TiO_2_ in the visible spectrum, while Anitha et al. [41] reported that the charge transportation could be eased in TiO_2_ due to additional bands, which can be achieved through cationic doping. Panepinto et al. [34] synthesized the p-type TiO_2_:N film with different nitrogen doping concentrations. As the N-doping increases, results show a change in light transmittance (%), optical bandgap (E_g_), Hall coefficient (cm^3^/C), carrier density (cm^−3^), conductivity (Ω^−1^·cm^−1^), and mobility (cm^2^·v^−1^·s^−1^). Additionally, a further increase in nitrogen concentration would lead to short-circuit in the cell, making it inappropriate as a photocathode. Previous studies inferred that TiO_2_ has a tuneable bandgap nature that depends on the nitrogen concentration [40,42].

### 2.3. Recipe of the PSC Structure

A formamidinium lead iodide (FAPbI_3_) active layer recipe was considered to investigate the performance, and the optimum high responsive match in power output was reported. Certain characteristics of the FA present in the absorber layer of PSCs include a lower bandgap of 1.48 eV, lower defect states, and better thermal stability compared to methylammonium (MA). Hence, FA-based PSCs are considered one of the most promising light-absorbing perovskite materials [43]. In view of accomplishing better cell stability, MAPbI_3_ absorber layer has gradually been replaced with FAPbI_3_ [44]. However, α-FAPbI_3_ tends to form an undesirable metastable non-perovskite phase transition and a thermodynamically stable photoinactive δ-polymorph state on exposure to ambient conditions. This causes a defects-induced non-ideal interfacial recombination leading to quicker cell failure [45,46,47]. Due to this fatal issue, the commercialization of FAPbI_3_-based perovskite solar cells is hampered. Therefore, nano-localization effects are one of the potential methods to stabilize the pure α-FAPbI_3_ phase. Table 1 reports the recent approaches that modern researchers have identified to stabilize the crystal structure of the pure α-FAPbI_3_ state.

The HTL medium used in an FA-based PSC structure is a novel concept in this research. The literature reports about the organic and inorganic-based HTLs to date, while oxide-based elements used as HTLs are still scarce. Recent adoptions in the HTL medium pertaining to formamidinium recipes are tabulated in Table 2. The adoption of TiO_2_:N as an HTL delivers an air-stable charge transport medium that was never performed earlier. Henceforth, the current study proposes simulating a planar n-i-p FA-based perovskite structure with TiO_2_:N as the HTL layer in the conventional TiO_2_-based ETL recipe.

The planned n-i-p PSC configuration structure is illustrated in Figure 1a. The idea of adopting TiO_2_ as HTL is a unique technique for developing a low-cost PSC. The literature reports that the bandgap and light transmission (%) is significantly reduced as the doping concentration increases. To develop a p-type epitaxial N-doped TiO_2_ thin film, Vasu et al. [38] optimized the bandgap of TiO_2_:N. The initial bandgap of bulk anatase n-type TiO_2_ is noted to be 3.23 eV; upon doping with 4.0% concentration (from XPS analysis) of nitrogen, the bandgap was reduced to 3.07 eV in the TiO_2_:N film, which also reported p-type behaviour due to the decrease in the fermi-energy levels. Preliminary simulations were performed in this research, and it was observed that the increase in nitrogen concentration would induce a short circuit when the HTL bandgap falls below 2.5 eV. Panepinto et al. [34] also reported a similar pattern when developing TiO_2_:N layers for dye-sensitized solar cells. Therefore, the doping concentration of nitrogen must be modulated to attain a bandgap of more than 2.5 eV for an ideal HTL medium with optimum light transmission properties. However, by considering previous experimental studies, this study investigates the behaviour of the PSC structure (FTO/TiO_2_/FAPbI_3_/TiO_2_:N) with two proposed HTL bandgaps of 3.0 eV and 2.5 eV.

The bandgap values of the corresponding layers in the proposed PSC structure are illustrated in Figure 1b.

The HTLs that pertain to nitrogen as a doping element result in two different bandgaps based on the doping concentration, i.e., a lower value of 2.5 eV for TiO_2_:N_a_ and a higher value of 3.0 eV for TiO_2_:N_b_. Consequently, the numerical simulation is performed with a combination of TiO_2_-based ETL and PSC active layer. SCAPS simulation for the proposed recipe is performed using distant parameters collected from various experimental and simulation studies, reported in Table 3. Based on HTL bandgaps, the two different recipes proposed for this study are:**Recipe-1**:FTO/TiO_2_/FAPbI_3_/TiO_2_:N_a_/Ag**Recipe-2**:FTO/TiO_2_/FAPbI_3_/TiO_2_:N_b_/Ag

**Table 3 molecules-27-07927-t003:** Simulation parameters of different layers within the PSC structure [34,38,63,64].

Parameter	FTO	TiO_2_	FAPbI_3_	TiO_2_:N_a_	TiO_2_:N_b_
Thickness ‘t’ (nm)	400	50	** *300 ** **	100	100
Band gap ‘E_g_’ (eV)	3.5	3.2	1.51	2.5	3.0
Electron affinity ‘χ’ (eV)	4	4	4	2.2	2.2
Dielectric Permittivity ‘ε_r_’	9	9	6.6	3	3
CB EDOS ‘N_c_’ (cm^−3^)	2.2 × 10^18^	2.1 × 10^18^	1.2 × 10^19^	1.3 × 10^18^	1.3 × 10^14^
VB EDOS ‘N_v_’ (cm^−3^)	2.2 × 10^18^	2.2 × 10^17^	1.2 × 10^19^	1.3 × 10^19^	1.3 × 10^15^
e^−^ thermal velocity (cm·s^−1^)	1 × 10^7^	1 × 10^7^	1 × 10^7^	1 × 10^7^	1 × 10^7^
h^+^ thermal velocity (cm·s^−1^)	1 × 10^7^	1 × 10^7^	1 × 10^7^	1 × 10^7^	1 × 10^7^
Electron mobility ‘μ_n_’ (cm^2^/V·s)	20	20	2.7	1.5	2.0
Hole mobility ‘μ_h_’ (cm^2^/V·s)	10	10	1.8	1.5	2.0
Shallow donor density ‘N._D_.’ (cm^−3^)	2 × 10^19^	9 × 10^16^	1.3 × 10^16^	0	0
Shallow Acceptor density ‘N._A_.’ (cm^−3^)	0	0	1.3 × 10^16^	1.3 × 10^19^	1.3 × 10^14^
Defect density ‘N_t_’ (cm^−3^)	10^15^	10^15^	** *1 × 10^13^ ** **	10^15^	10^15^

* Varied parameter.

Further investigations are conducted to examine the cell performance and behaviour concerning modulating the absorber layer thickness and defect density. The absorber layer thickness (∆t) is varied from 300 nm onwards. In contrast, the defect density of the perovskite absorber layer is varied (∆N_t_) for four different attributes from 1 × 10^13^ to 1 × 10^16^ cm^−3^, respectively. The simulations in this current study were performed by considering specific assumptions.

(a)The phase of the formamidinium crystal structure is stable at the α-phase; there is no drift into the δ-phase.(b)The temperature coefficient on the perovskite recipes is precluded.

## 3. Results and Discussion

Considering the electrical parameters in Table 3, and no resistances, numerical simulations were performed to estimate the performance output for the proposed recipes from the lowest absorber layer thickness of 300 nm. Results attained through SCAPS simulation are presented in Table 4. The current-voltage characteristics (J-V) pertaining to the recipes are displayed in Figure 2a.

The results infer the significance of N-doped TiO_2_ as an HTL with reasonable PCE attained from both recipes. However, it is observed that the TiO_2_:N_b_ HTL with a bandgap of 3.0 eV has attained higher output when compared to the TiO_2_:N_a_ HTL with a bandgap of 2.5 eV. Interestingly, a minimal change in J_sc_ and V_oc_ was observed for the recipes. With the increase in HTL bandgap from 2.5 eV to 3.0 eV, the fill factor (FF, %) increased by 11.61%, which is reflected in the rise of power conversion efficiency (%) by 3.12%. The enhancement of performance with the increase in HTL bandgap can be inferred from the bandgap grading that induces efficient hole transport. This analysis can theoretically support the combination ideology of TiO_2_/FAPbI_3_/TiO_2_:N as a suitable PSC structure that can attain considerable power outputs on par with organic and inorganic HTLs. The patterns depicted indicate the wider area of the TiO_2_:N_b_ curve that specifies a higher power output than the TiO_2_:N_a_ curve. From the quantum efficiency spectrum depicted in Figure 2b, it is evident that the FAPbI_3_ recipe encounters a similar profile for different HTL bandgaps. However, recipe-1 reported a slightly higher output between 375 nm to 500 nm but was surpassed by recipe-2 by the end.

### 3.1. Effect of Perovskite Absorber Layer Thickness

In this section, a numerical analysis is carried out to investigate the performance output for the designed recipes when the perovskite absorber layer thickness is varied. The PCE is one of the vital factors that is expected to invariably respond to the degree of change in absorber layer thickness. Hence, this attempt would project the optimum and maximum thickness of the absorber layer to yield better output.

Conceptually, the increase in the absorber layer thickness influences the performance parameters of a PSC. With the increase in thickness, the short circuit current (J_sc_, mA/cm^2^) tends to increase since it is attributed to more electron-hole pairs in the absorber layer, whereas the open circuit voltage (V_oc_, V) decreases due to the increment in the dark saturation current that increases the recombination of charge carriers. Seemingly, the fill factor (FF, %) holds an inversely proportional relationship with an increase in thickness due to the increase in series resistance and internal power dissipation. Additionally, the increase in J_sc_ and a drop in FF would reflect in the increase of device performance due to the balanced charge transport [5]. Lastly, the power conversion efficiency (PCE, %) tends to increase with thickness but decreases beyond a saturation level (maximum diffusion length). Beyond the saturation point, the fill factor is reported to drop due sheet resistance of the active layer, which is a material perspective phenomenon.

These numerical simulations reveal that the photovoltaic parameters report a change when the layer thickness varies. The primary electrical parameters from Table 3 are considered while the thickness is varied proportionally from 300 nm until the saturation level and maximum allowable thickness is observed. The simulation results show that both recipes’ performance profile has a divergent pattern. The point of convergence for the FTO/FAPbI_3_/TiO_2_:N_a_ recipe with 2.5 eV HTL bandgap is below 1.1 V. The FTO/FAPbI_3_/TiO_2_:N_b_ recipe with 3.0 eV HTL bandgap reported the convergence point at 1.2 V, thereby depicting the influence of the HTL bandgap on the cell performance.

Figure 3a,b illustrate the J-V characteristic curve for both recipes based on different HTL bandgaps. It shows that the influence of perovskite absorber layer thickness gradient (∆t) on the performance output is apparent. For both recipes, the absorber layer thickness had a high initial power output that was observed to reduce gradually as the voltage and current increased. As ∆t increased, the PCE (%) was observed to show an incremental pattern. The increase in absorber thickness will enhance electron/hole pair generation and electron mobility. Indeed, the peak of the absorber thickness corresponding to the saturation point is 600 nm FTO/FAPbI_3_/TiO_2_:N_a_ recipe and 1000 nm for the FTO/FAPbI_3_/TiO_2_:N_b_ recipe, combinedly depicted in Figure 4, respectively. The patterns obtained in Figure 4 depict a proportionate behaviour among the power conversion efficiency and absorber layer thickness. The increase in HTL bandgap from 2.5 eV to 3.0 eV has fostered the ability to increase the perovskite absorber layer thickness to 1000 nm in recipe-2 before the saturation point can be observed. This is the point beyond which the thermal recombination happens, and the PCE gradually tends to decline. Increasing the absorber layer thickness to up to 1000 nm can attain a higher PCE of up to 26.99% for the corresponding recipe. The performance characteristic attained due to the gradient in ∆t is tabulated in Table 5 and the relation between J_sc_, V_oc_, FF and PCE are illustrated in Figure 5a,b. Therefore, this simulation identifies the peak threshold absorber layer thickness for the FAPbI_3_ recipes that employ TiO_2_:N as the HTL and TiO_2_ as the ETL, respectively.

### 3.2. Effect of Defect Density in the Perovskite Absorber Layer

The quality and structure of the PSC absorber layer play a significant role in delivering an efficient power output. The defect density (N_t_) of the absorber layer influences the performance parameters as the film quality deteriorates. This phenomenon causes a density trap and rise in the recombination of charge carriers, which reflects on the cell output [65]. Madan et al. [66] described that the performance of the PSC is directly influenced by the defect densities on both the perovskite/ETL and the perovskite/HTL. The effect is more intense when light is illuminated from the ETL side due to the high rate of photons being absorbed near the perovskite/ETL interface. This research investigates and reports the impact of defect density from the perovskite/ETL side as the illumination is projected from the ETL side. Hence, this simulation studies the performance of the proposed perovskite recipes with a gradient in defect density (∆N_t_) from 1 × 10^13^ to 1 × 10^16^ cm^−3^ for the lowest ∆t of 300 nm and the saturation peak ∆t value of 600 nm for recipe-1 and 1000 nm for recipe-2. The J-V characteristics for both recipes are reported in Figure 6a,b. Furthermore, the simulation results notify a consequent drop in the performance output parameters, as reported in Table 6. The simulation results for both recipes infer the impact of the ∆N_t_ on the PCE (%). With the increase in defect density by 1 × 10^1^ cm^−3^, the J_sc_, FF (%), and V_oc_ were observed to reduce logarithmically, resulting in a drop in the PCE (%) in both the recipes. However, it is observed that the increase in defect density has a lesser influence on the J_sc_, but the V_oc_ was observed to respond in proportional change. The FF (%) in recipe-1 was noted to drop drastically with the increased defect density for both the ∆t gradients. Interestingly, though the defect density in recipe-2 increased to 1 × 10^16^ cm^−3^, the FF (%) for both the ∆t gradients was attributed to being 54.16% and 44.63%, resulting in the PCE of 11% and 9.17%, which is nearly 1.7 times the PCE attained in recipe-1, respectively. Additionally, it is identified that a reasonable PCE (%) of nearly 16% is attained even if the defect density is increased to 1 × 10^15^ cm^−3^ in recipe-2. This output infers the role of the HTL bandgap that increased from 2.5 eV to 3.0 eV in the proposed recipes.

### 3.3. Comparison of Different Polymeric HTLs Used in an FA-Based PSC Recipe

From the previous sessions, it is evident that the HTL derived from TiO_2_:N has the potential to deliver adequate electrical performance. Nitrogen doping on c-TiO_2_ corresponds to developing different bandgaps, which play a functional role in the cell performance. The literature reports (Table 2) that polymeric HTLs such as Spiro-OMeTAD, P3HT, and PEDOT: PSS are most used and viable to feed the role of charge transport mediums in an FA-based cell structure. However, the intrinsic instability, high cost [18,19,67], and multi-step synthesis of these polymeric layers still creates a vacuum for alternative HTLs to be developed.

From the gist of the previous analysis, this further session studies and numerically compares the cell performance output between TiO_2_:N and polymeric HTLs. The electrical parameters from Table 3 are considered, focusing on attaining a better result with the PSC absorber layer thickness of 600 nm and higher TiO_2_:N bandgap of 3.0 eV. The polymeric layers’ parameters are considered from the literature in Table 2, which collectively reported simulation and experimental results. Table 7 tabulates the simulative electrical parameters pertaining to this analysis. The J-V curve obtained from numerical simulation is pictured in Figure 7, while the corresponding output performance parameters are tabulated in Table 8.

The performance parameters and the J-V patterns explain the positive potential that is attainable with TiO_2_:N when used as an HTL in an FTO/TiO_2_/FAPbI_3_/HTL/Ag recipe. The magnified view in Figure 7 reports the drift in the short circuit current for the respective HTLs. TiO_2_:N resulted in a PCE rise of 1.03% and 1.15% over Spiro-OMeTAD and P3HT. It reported a similar trend with 0.9% less PCE against PEDOT: PSS. The J_sc_ for these recipes was observed to be similar with minor variation, whereas the fill factor was seen to modulate, reflected in the final PCE.

## 4. Conclusions

In this present study, a formamidinium lead iodide (FAPbI_3_) perovskite solar cell was optimized with a novel low-cost HTL in the form of nitrogen-doped titanium dioxide (TiO_2_:N). The performance output influenced by two different HTL bandgaps (2.5 eV and 3.0 eV) was investigated and significantly compared through SCAPS simulation. In addition, the effect of modulation in the absorber layer thickness (∆_t_) and defect density (∆N_t_) was studied, with optimum results reported. The increase in perovskite absorber thickness and increase in HTL bandgap has witnessed a rise in the PCE (%), reaching nearly 26.99%, an exceptional output for a low-cost FAPbI_3_ solar cell. Though the defect density in the absorber layer increased, the recipe with a high HTL bandgap achieved a reasonable bandgap of 16.59%. Lastly, the behaviour of this HTL was compared with three other polymeric mediums on a standard FA recipe. The simulation results denote a PCE of 26.34%, which is compatibly higher than Spiro-OMeTAD and P3HT. However, the practical feasibility of TiO_2_:N as a low-cost and stable HTL is undetermined until experimentation is performed, which is considered the future work in this research. Based on simulations, these results conclude that TiO_2_:N is a suitable HTL that could attain a PCE (%) equivalent to organic and inorganic HTLs. Additionally, with the rendering optical properties in N-doped TiO_2_, these recipes could aid in developing semi-transparent perovskite cells suited for various applications. Based on the insights gained from this research, this work provides preliminary ways to lower the cost of an FA-based perovskite structure that can be developed soon. Finally, by considering the approaches in Table 2, the viability of different techniques is feasible to confront the phase stability of the α-FAPbI_3_ structure.

## Figures and Tables

**Figure 1 molecules-27-07927-f001:**
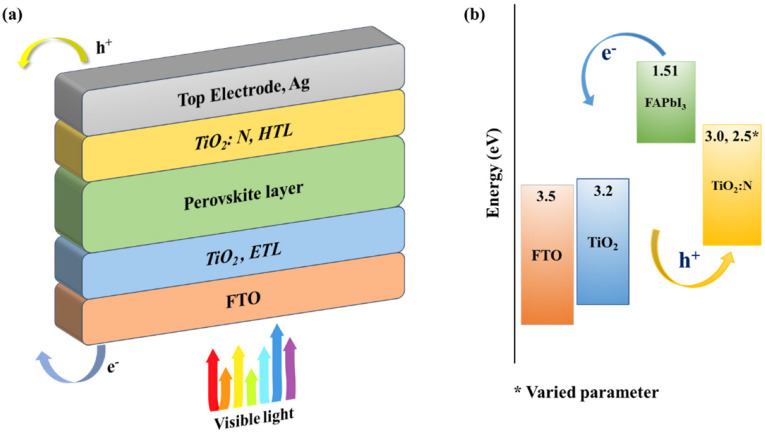
(**a**) Structure of the PSC with TiO_2_ as ETL and TiO_2_:N as HTL; (**b**) energy band diagram of the PSC layers with FAPbI_3_ absorber layer and different bandgap HTLs.

**Figure 2 molecules-27-07927-f002:**
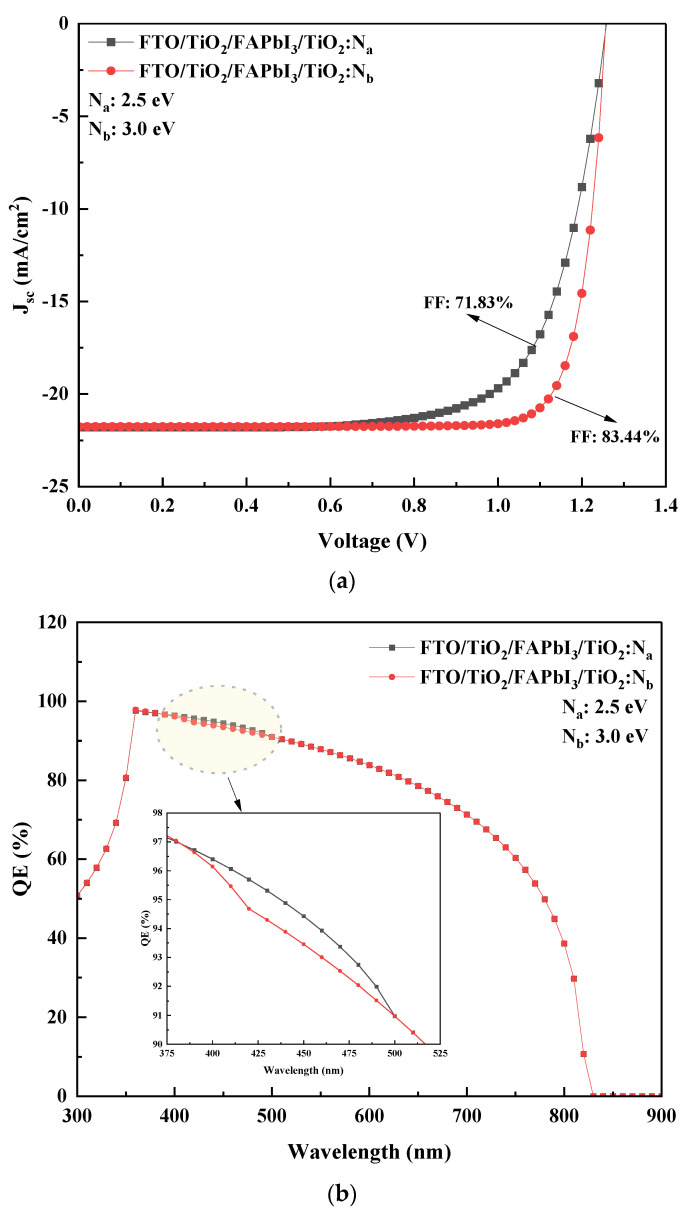
(**a**) Simulated J-V characteristics for the two PSC recipes. (**b**) Quantum efficiency spectrum of the simulated PSC recipes.

**Figure 3 molecules-27-07927-f003:**
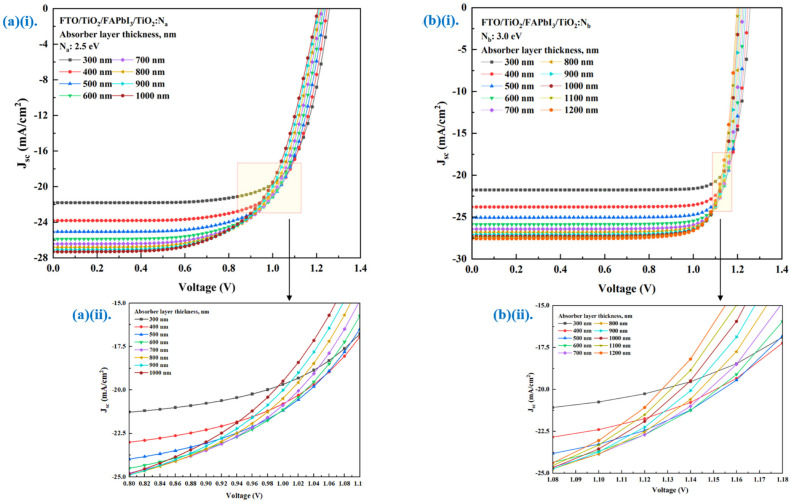
(**a**) (**i**) J-V characteristic curve for the FTO/FAPbI_3_/TiO_2_:N_a_ recipe with ∆t; (**ii**) a magnified view of the convergence point from the applied voltage range 0.8–1.1 V. (**b**) (**i**) J-V characteristic curve for the FTO/FAPbI_3_/TiO_2_:N_b_ recipe with ∆t; (**ii**) a magnified view of the convergence point in the applied voltage range 1.08–1.18 V.

**Figure 4 molecules-27-07927-f004:**
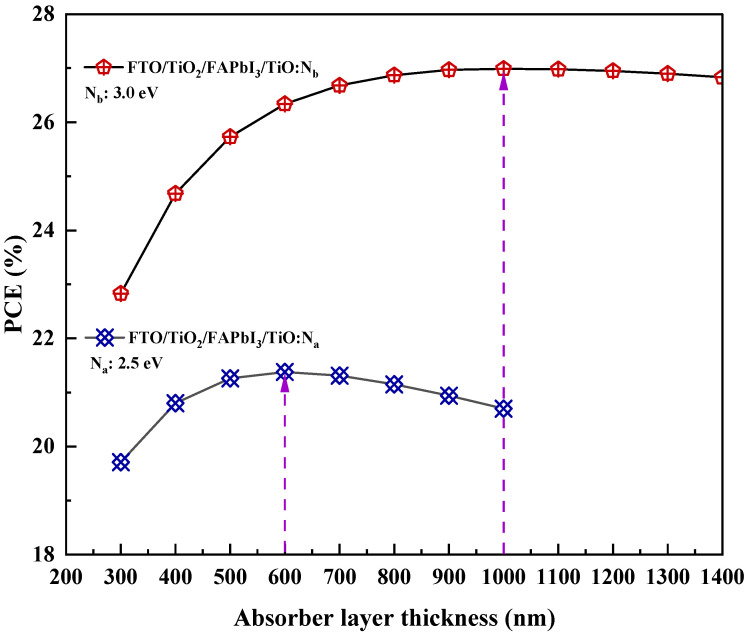
Spectrum of the maximum allowable absorber layer thickness for both recipes.

**Figure 5 molecules-27-07927-f005:**
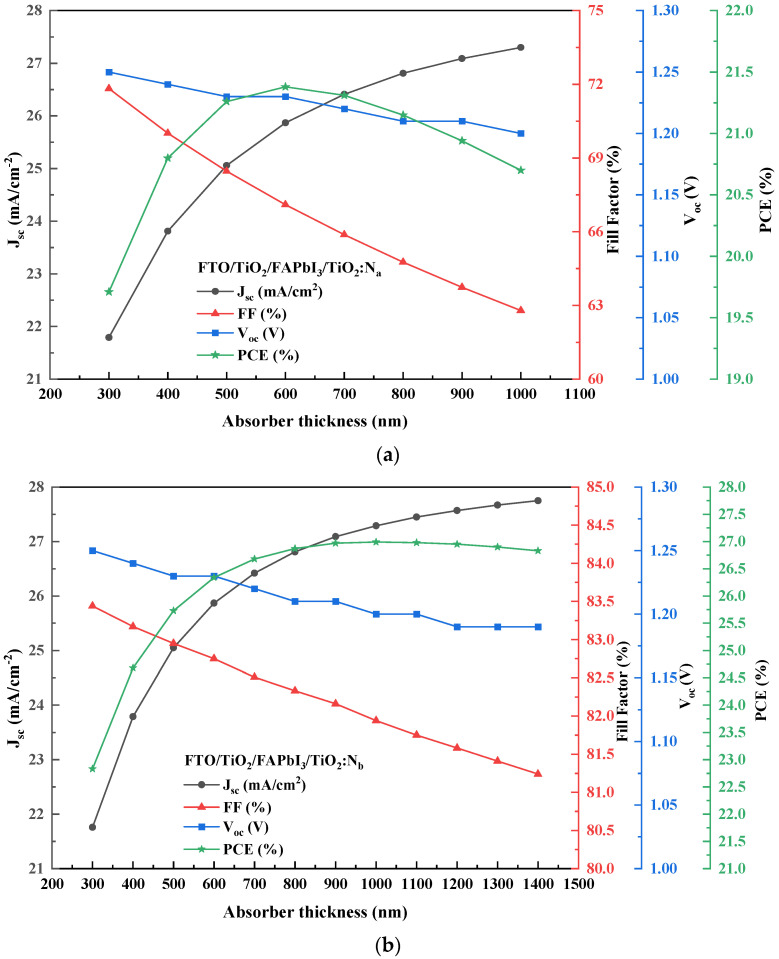
(**a**) Performance parameter curves for the FTO/FAPbI_3_/TiO_2_:N_a_ recipe. (**b**) Performanc parameter curves for the FTO/FAPbI_3_/TiO_2_:N_b_ recipe.

**Figure 6 molecules-27-07927-f006:**
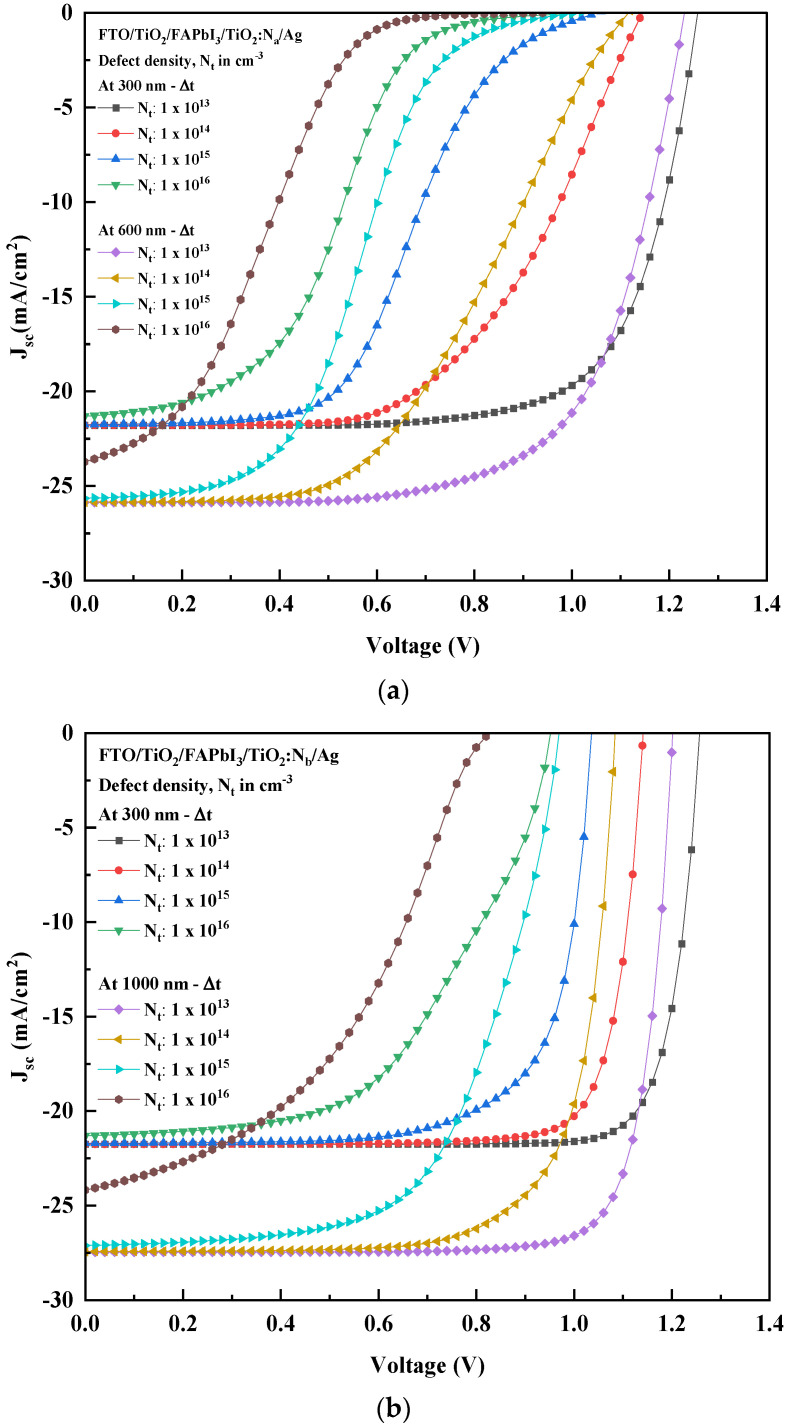
(**a**) J-V curve for the FTO/FAPbI_3_/TiO_2_:N_a_ recipe with ∆N_t_. (**b**) J-V curve for the FTO/FAPbI_3_/TiO_2_:N_b_ recipe with ∆N_t_.

**Figure 7 molecules-27-07927-f007:**
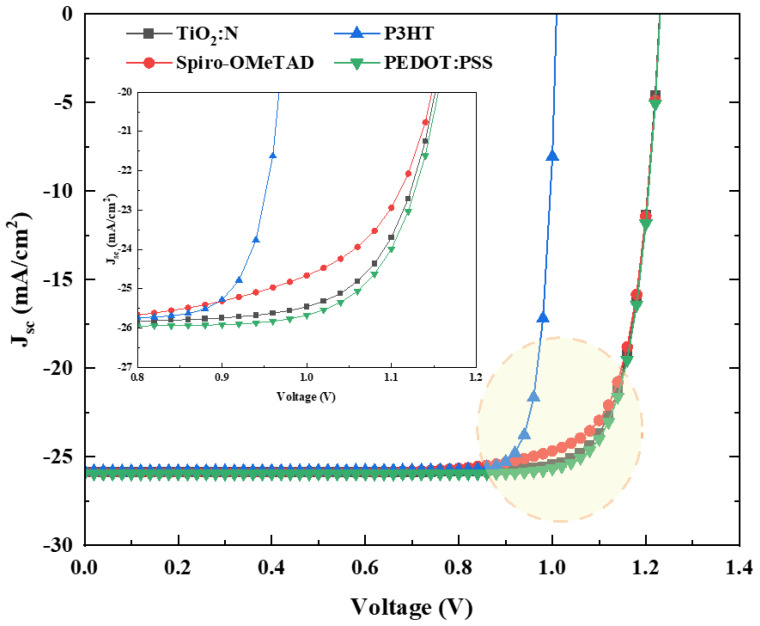
J-V characteristic curve for the HTL comparative analysis with a magnified view that depicts the drift patterns between 0.8–1.2 V.

**Table 1 molecules-27-07927-t001:** Recent research that reports the techniques to stabilize the pure α-FAPbI_3_ state.

S. No.	Year	Author [Reference]	Recipe	Remarks
1	2022	Wang et al. [46]	SnO_2_/FAPSC/Spiro-OMeTAD	4-fluorophenylmethylammonium iodide (F-PMAI) was used to modulate surface structure and energy level alignment.
2	2022	Kundu et al. [47]	FAPbI_3_ crystals	The α-FAPbI_3_ single crystals are stabilized through Pb-site doping with a heterovalent metal–bismuth (Bi). The optimum concentration of Bi extends the phase change by four orders of magnitude.
3	2022	Bu et al. [48]	(FA-Cs) lead halide	Controlled the formation of intermediate phases during the growth of formamidinium–caesium lead triiodide perovskite films by using methylammonium chloride additives in the co-solvent system of N-methyl-2-pyrrolidone/N, N-dimethylformamide.
4	2022	Liu et al. [49]	FA-Cs + NiO_x_ (HTL)	A molecular additive—the organic dye coumarin 343—was used to reduce V_oc_ loss and improve efficiency.
5	2022	Li et al. [50]	(FA-Cs) lead halide	Additives reduced crystallization and defects of the FA-Cs perovskite film.

**Table 2 molecules-27-07927-t002:** Recent simulations that report various organic and inorganic HTLs adopted in an FA-based PSC structure (2022–2021).

S. No.	Year	Author	Perovskite Solar Cell Structure (ETL/Absorber/HTL)	Performance Factors (V_oc_ (V), J_sc_ (mA/cm^2^), FF (%), PCE (%)	Remarks	Reference
1	2022	Vishnuwaran et al.	TiO_2_/FASnI_3_/CuO_2_	V_oc_: 0.7921, J_sc_: 29.61, FF: 78.14, PCE: 18.10	Varied absorber layer thickness—350 nm yielded the best results	[51]
2	2022	Niloy et al.	SnO_2_/FA_0.83_Cs_0.17_PbI_0.5_Br_2.5_/MoOx	PCE up to 22.89% for absorber thickness of 169 nm	Absorber layer thickness influences the PCE	[52]
3	2022	Sabbah et al.	(TiO_2,_ ZnOS)/FA_1−x_Cs_x_SnI_3_/CuO_2_	V_oc_: 0.89, J_sc_: 31.4, FF: 78.7, PCE: 22	ZnOS exhibited stable behaviour and was better than TiO_2_	[53]
4	2022	Vishnuwaran	ZnOS/FASnI_3_/CuI	V_oc_: 6.20, J_sc_: 30.77, FF: 12.68, PCE: 24.22	CuI and ZnOS are considered ideal replacements for Spiro-OMeTAD and TiO_2_	[54]
5	2022	Teimouri et al.	Cs_0.05_ (FA_x_MA_[1−x]_)_0.95_Pb(I_0.83_Br_0.17_)_3_	Attained a PCE of up to 20.98%	Analyzed bandgap ratios between 2.175 eV to 1.5 eV and x factor influenced the power output	[55]
6	2021	Jannat et al.	SnO_2_/FA_0.83_Cs_0.17_ PbI_1.5_Br_1.5_/MoO_x_	V_oc_: 1.44, J_sc_: 17.04, FF: 81.83, PCE: 20.10	MoO_x_ exhibited a low valence band offset. HTL and absorber thickness was varied	[56]
7	2021	Stanić et al.	TiO_2_/Rb_0.05_Cs_0.1_FA_0.85_PbI_3_/Spiro-OMeTAD	V_oc_: 0.80, J_sc_: 20.60, FF: 45.51, PCE: 7.35	Absorber layer thickness, defect density concentration, and the influence of the resistivity were analyzed	[57]
8	2021	Alipour and Ghadimi	PC_61_BM/FASnI_3_/PEDOT: PSS+WO_3_	Vo_c_: 1.12, J_sc_: 24.65, FF: 86.02, PCE: 23.69	FA depicted better outputs that MA based structures	[58]
9	2021	Tara et al.	Zn (O_0.3_, S_0.7_)/FASnI_3_/CuSCN	V_oc_: 1.08, J_sc_: 28.12, FF: 84.96, PCE: 25.94	Variations in electron affinity, CBO, doping density, and thickness of Zn (O_0.3_, S_0.7_) were analyzed	[59]
10	2021	Patil et al.	ZnO/FAPbI_3_/Spiro-OMeTAD	V_oc_: 0.99, J_sc_: 26.75, FF: 79.80, PCE: 21.26	FA-based PSC structures have depicted higher efficiency than MA	[60]
11	2021	Kanoun et al.	TiO_2_/FAPbI_3_/PTAA and Cu_2_O	PCE of up to 24% for absorber thickness of 400 nm	PTAA and Cu_2_O as HTLs enhance change carriers	[61]
12	2021	Bhardwaj et al.	SnO_2_/FA_0.85_Cs_0.15_Pb(I_0.85_Br_0.15_)_3_/Spiro-OMeTAD and Cu_2_O	Spiro-OMeTAD (HTL)—PCE: 15.36% Cuprous oxide (HTL)—PCE: 19.38%	CuO_2_ delivers the highest efficiency when compared to other inorganic HTLs	[62]

**Table 4 molecules-27-07927-t004:** Summarised photovoltaic parameters of the two recipes from SCAPS simulation.

PSC Recipe	J_sc_ (mA/cm^−2^)	FF (%)	V_oc_ (V)	PCE (%)
FTO/TiO_2_/FAPbI_3_/TiO_2_:N_a_/Ag	21.798	71.83	1.25	19.71
FTO/TiO_2_/FAPbI_3_/TiO_2_:N_b_/Ag	21.760	83.44	1.25	22.83

**Table 5 molecules-27-07927-t005:** Performance parameters for both recipes with varied ∆t.

Recipe	∆t (nm)	J_sc_ (mA/cm^−2^)	FF (%)	V_oc_ (V)	PCE (%)
FTO/TiO_2_/FAPbI_3_/ TiO_2_:N_a_/Ag	300	21.79	71.83	1.25	19.71
	400	23.81	70.01	1.24	20.80
	500	25.06	68.46	1.23	21.26
	**600 ^#^**	**25.87**	**67.10**	**1.23**	**21.38**
	700	26.41	65.88	1.22	21.31
	800	26.81	64.76	1.21	21.15
	900	27.09	63.74	1.21	20.94
	1000	27.30	62.79	1.20	20.70
FTO/TiO_2_/FAPbI_3_/ TiO_2_:N_b_/Ag	300	21.76	83.44	1.25	22.83
	400	23.79	83.17	1.24	24.68
	500	25.05	82.95	1.23	25.73
	600	25.87	82.75	1.23	26.34
	700	26.42	82.51	1.22	26.68
	800	26.81	82.33	1.21	26.87
	900	27.09	82.16	1.21	26.97
	**1000 ^#^**	**27.29**	**81.94**	**1.20**	**26.99**
	1100	27.45	81.75	1.20	26.98
	1200	27.57	81.58	1.19	26.95
	1300	27.67	81.41	1.19	26.90
	1400	27.75	81.24	1.19	26.83

^#^ Saturation point.

**Table 6 molecules-27-07927-t006:** Performance parameters for proposed recipes with ∆N_t_ and ∆t.

Recipe	Absorber Layer Thickness (∆t)	Defect Density (N_t_)	J_sc_ (mA/cm^−2^)	FF (%)	V_oc_ (V)	PCE (%)
FTO/TiO_2_/FAPbI_3_/ TiO_2_:N_a_/Ag	**300 nm**	1 × 10^13^	21.79	71.83	1.25	19.71
		1 × 10^14^	21.79	55.15	1.14	13.92
		1 × 10^15^	21.75	45.68	1.05	10.44
		1 × 10^16^	21.31	33.12	0.99	7.06
	**600 nm ^#^**	1 × 10^13^	25.87	67.10	1.23	21.38
		1 × 10^14^	25.85	48.79	1.11	14.08
		1 × 10^15^	26.62	37.12	1.01	9.62
		1 × 10^16^	23.72	22.17	0.93	4.94
FTO/TiO_2_/FAPbI_3_/ TiO_2_:N_b_/Ag	**300 nm**	1 × 10^13^	21.76	83.44	1.25	22.83
		1 × 10^14^	21.75	81.76	1.14	20.31
		1 × 10^15^	21.71	72.57	1.03	16.33
		1 × 10^16^	21.31	54.16	0.95	11.00
	**1000 nm ^#^**	1 × 10^13^	27.29	81.94	1.20	26.99
		1 × 10^14^	27.27	74.82	1.08	22.22
		1 × 10^15^	26.99	63.02	0.97	16.59
		1 × 10^16^	24.45	44.63	0.84	9.17

^#^ Saturation point.

**Table 7 molecules-27-07927-t007:** Simulation parameters for TiO_2_:N and Polymeric HTLs.

Parameters	Substrate	ETL	Perovskite Absorber Layer	Novel HTL	Polymeric HTLs		
	FTO	TiO_2_	FAPbI_3_	TiO_2_:N	Spiro-OMeTAD [62]	P3HT [68]	PEDOT: PSS [58]
Thickness ‘t’ (nm)	400	50	600	100	200	50	200
Band gap ‘E_g_’ (eV)	3.5	3.2	1.51	3.0	2.88	1.1	1.8
Electron affinity ‘χ’ (eV)	4	4	4	2.2	2.05	4.6	3.4
Dielectric Permittivity ‘ε_r_’	9	9	6.6	3	3	13.6	18
CB EDOS ‘N_c_’ (cm^−3^)	2.2 × 10^18^	2.1 × 10^18^	1.2 × 10^19^	1.3 × 10^14^	2.2 × 10^18^	3 × 10^18^	2.2 × 10^18^
VB EDOS ‘N_v_’ (cm^−3^)	2.2 × 10^18^	2.2 × 10^17^	1.2 × 10^19^	1.3 × 10^15^	1.8 × 10^19^	2 × 10^19^	1.8 × 10^19^
e^−^ thermal velocity (cm·s^−1^)	1 × 10^7^	1 × 10^7^	1 × 10^7^	1 × 10^7^	1 × 10^7^	1 × 10^7^	1 × 10^7^
h^+^ thermal velocity (cm·s^−1^)	1 × 10^7^	1 × 10^7^	1 × 10^7^	1 × 10^7^	1 × 10^7^	1 × 10^7^	1 × 10^7^
Electron mobility ‘μ_n_’ (cm^2^/V·s)	20	20	2.7	2	2 × 10^−4^	25	4.5 × 10^−2^
Hole mobility ‘μ_h_’ (cm^2^/V·s)	10	10	1.8	2	2 × 10^−4^	25	4.5 × 10^−2^
Shallow donor density ‘N._D_.’ (cm^−3^)	2 × 10^19^	9 × 10^16^	1.3 × 10^16^	0	0	0	0
Shallow Acceptor density ‘N._A_.’ (cm^−3^)	0	0	1.3 × 10^16^	1.3 × 10^14^	2 × 10^19^	3 × 10^16^	1 × 10^20^
Defect density ‘N_t_’ (cm^−3^)	1 × 10^15^	1 × 10^15^	1 × 10^13^	1 × 10^15^	1 × 10^15^	1 × 10^13^	1 × 10^15^

**Table 8 molecules-27-07927-t008:** Performance parameters attained from the HTL comparative analysis.

PSC Recipe	J_sc_ (mA/cm^−2^)	FF (%)	V_oc_ (V)	PCE (%)
FTO/TiO_2_/FAPbI_3_/TiO_2_:N/Ag	25.87	82.75	1.23	26.34
FTO/TiO_2_/FAPbI_3_/Spiro-OMeTAD/Ag	25.86	79.83	1.23	25.42
FTO/TiO_2_/FAPbI_3_/P3HT/Ag	25.81	87.62	1.01	22.85
FTO/TiO_2_/FAPbI_3_/PEDOT: PSS/Ag	25.95	83.30	1.23	26.61

## Data Availability

Not applicable.

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
