# Peer review of "Nitrogen-Doped Titanium Dioxide as a Hole Transport Layer for High-Efficiency Formamidinium Perovskite Solar Cells"

_molecules, 2022, doi:10.3390/molecules27227927_

Round 1
Reviewer 1 Report
See the attachment

Author Response
- The development of the solar cell is very fast, the introduction section should be modified though citing recent references related studies to let readers know more clearly about the photovoltaic field. For example: Energy Technol. 2022, 10, 2200504 and Opt. Mater. 2022, 129,112520,
Response: The authors would like to thank the reviewer for the valuable suggestion to improve the manuscript. An introduction about importance of solar energy and photovoltaics has been added in the revised manuscript. [Page:02, Line:35–49]. Accordingly, the recommended papers are also referred in the revised manuscript [References: 1,5].
- The author says: "... this concept would foster to the feasibility of developing stable and semi-transparent PSCs with a low-cost HTL. ", "…perovskite solar cell was optimised with a novel low-cost HTL in the form of nitrogen-doped Titanium dioxide (TiO2:N). ", " … the practical feasibility of TiO2:N as a low-cost and stable HTL is undetermined until experimentation is performed…" and "…this work provides preliminary ways to low down the cost and improve the stability in a formamidinium based perovskite structure that can be developed in near future." How do you get the reason why TiO2:N costs less than PEDOT:PSS or Spiro- OMeTAD HTL? In addition, the stability of perovskite solar cell with TiO2:N HTL also needs the support of experimental data. I think the description of these aspects should be deleted or supported by references
Response: The authors would like to thank the reviewer for the comment. This work comprises of numerically simulating an FA based perovskite structure with a HTL in the form of TiO2:N. From literature [18,19,68], it is relevant that polymeric HTLs such as Spiro-OMeTAD and PEDOT: PSS are unstable and expensive which has fostered researchers to switch to inorganic HTLs. Henceforth, we have proposed a material that can be developed with a competitively lower cost. Our argument is also based on the commercially available data on the prices of these materials (Spiro-OMeTAD - 700$, PEDOT:PSS – 240 $ and TiO2 – 45 $, https://www.sigmaaldrich.com/IN/en) as well as the published data on the atmospheric stability and their influence on the stability of the PSCs.
Secondly, we agree with the reviewer in the aspect of stability of the proposed cell structure is obscure until further detailed experiments are performed. Henceforth, specific content on achieving stability (in the conclusion part) has been removed in the revised manuscript.
Reviewer 2 Report
The authors have discussed titanium dioxide in perovskite solar cells. Their attempt at a physical explanation is incomplete. The manuscript lacks any reference to physical length when discussing energy bands or applied voltages. This makes it impossible to determine the device current due to lack of information about drift and electric fields. I cannot trust figures containing device current, i.e. Figs. 2-7. Almost all data are simulation results, and the lack of systematic calibration to experimental data is a fatal mistake.
1. Fig. 1 misuses energy band diagrams
Fig. 1b shows flat band energy bandgaps and offsets for various materials in the perovskite solar cell. This is not appropriate for a working solar cell device. As soon as two materials are connected, a barrier region forms because of carrier exchange between them. This is the central mechanism to a working pn junction. It has formed the basis of semiconductor physics for over 70 years, and cannot be ignored. Furthermore, the flat band condition assumed in Fig. 1b is nearly impossible due to surface states, which the authors have also neglected.
2. Fig. 2 only shows illuminated cases, no dark current
pn junction mechanism is unconfirmed without zero illumination I-V. This means the solar cell action may or may not be happening where the authors expect.
3. Conversion to device current missing, should be nontrivial
With discussion for contacts or length scales missing, any conversion to device current cannot be verified.
4. Perovskite doping, direct/indirect missing
No semiconductor properties for perovskite discussed, meaning the photon absorption mechanism is completely missing.
5. Missing discussion of device contacts
This is one possible explanation for I-V and illuminated characteristics that the authors never ruled out.
Again, Fig. 2 – 7 report simulated device current, but there is no systematic calibration to experimental data. I cannot accept device current data in this manuscript without systematic calibration. The device discussion is severely incomplete, and further revisions will not be enough to fill in the gaps. This paper must be rejected.
Author Response
1. The authors have discussed titanium dioxide in perovskite solar cells. Their attempt at a physical explanation is incomplete. The manuscript lacks any reference to physical length when discussing energy bands or applied voltages. This makes it impossible to determine the device current due to lack of information about drift and electric fields. I cannot trust figures containing device current, i.e. Figs. 2-7. Almost all data are simulation results, and the lack of systematic calibration to experimental data is a fatal mistake.
Fig. 1 misuses energy band diagrams (thank the reviewers for the comment)
Response: The authors would like to thank the reviewer for the critical comment.
- We agree that the results presented in this paper is via simulation and that no experiment is conducted. However, in order to ensure the accuracy of claims, we have simulated the data of two well-known hole transport layers (Spiro-OMeTAD and PEDOT:PSS) and validated the simulation results with the experimentally reported data in open literature. Also, the simulation procedure for all the materials including the title compound (TiO2:N) kept exactly similar such that the uncertainties in the claims of the present manuscript could be minimized. Therefore, we believe that the data presented in this manuscript is consistent with the experimental data. Moreover, we employed the state-of-the-art level of the theory in the present work.
- Secondly, the authors thank the reviewer for the indicating the lapse projected in Fig.1 (Energy band diagram). We notify that the band energy diagram has been edited in the revised manuscript.
- 1b shows flat band energy bandgaps and offsets for various materials in the perovskite solar cell. This is not appropriate for a working solar cell device. As soon as two materials are connected, a barrier region forms because of carrier exchange between them. This is the central mechanism to a working p-n junction. It has formed the basis of semiconductor physics for over 70 years and cannot be ignored. Furthermore, the flat band condition assumed in Fig. 1b is nearly impossible due to surface states, which the authors have also neglected.
Fig. 2 only shows illuminated cases, no dark current
p-n junction mechanism is unconfirmed without zero illumination I-V. This means the solar cell action may or may not be happening where the authors expect.
Response: The authors would like to thank the reviewer for the comments. This simulation has been carried out with TiO2:N HTL having two band gaps viz., 2.5 eV and 3.0 eV (Bandgaps attained due to different levels of nitrogen doping – reported from experimental literature [50,34]. Fig 1.b. separately reports the HTLs with significant band gaps of 2.5eV and 3.0 eV. This representation has led to a misunderstanding that two HTLs are used in the structure. Which is not true! Henceforth, the authors wish to report that have Fig 1.b. is modified in the revised manuscript, to avoid further misconception.
Secondly, the authors wish to notify that solar cell simulations work only under illumination of light, which is the prime reason that only illumination is reported in this study. Dark current simulations are carried out in photodetectors; hence we have not considered simulations in this work.
- Conversion to device current missing, should be nontrivial
With discussion for contacts or length scales missing, any conversion to device current cannot be verified.
Response: The authors would like to thank the reviewer for the comment. Non-ideally, SCAPS simulation assumes a z++ero contact resistance and considers only the intrinsic electrical properties of the materials used in the structure. This limitation has been cited in the manuscript.
- Perovskite doping, direct/indirect missing
No semiconductor properties for perovskite discussed, meaning the photon absorption mechanism is completely missing.
Response: The authors would like to thank the reviewer for the comment. The authors notify that, we do not consider those aspects of doping in perovskite layer. We have used the intrinsic properties of FA based perovskite in our simulation. The idea of doping has been carried out in the HTL basis.
- Missing discussion of device contacts
This is one possible explanation for I-V and illuminated characteristics that the authors never ruled out.
Again, Fig. 2 – 7 report simulated device current, but there is no systematic calibration to experimental data. I cannot accept device current data in this manuscript without systematic calibration. The device discussion is severely incomplete, and further revisions will not be enough to fill in the gaps. This paper must be rejected.
Response: The authors would like to thank the reviewer for the comment.
As stated above, SCAPS simulation assumes a Zero contact resistance and considers only the intrinsic electrical properties of the materials used in the structure. The purpose of this manuscript is to scrutinize the relative advantages of a cheaper earth abundant metal oxide semiconductor, namely nitrogen doped titanium dioxide as a replacement for air-sensitive and expensive polymeric hole conducting materials. Our simulations are carried out with FTO and Ag as bottom and top contact. These points were discussed in the main manuscript.
Reviewer 3 Report
I find the styudy relevant for the perovskite research community encompassing solar cells and LEDs, etc. I have a few comments on the results and discussion as noted below which need to be addressed.
1. If possible, provide a schematic band-diagram and explain how the large NiO bandgap can enhance the PCE, considering the bandgap changes from 2.5 eV to 3 eV
2. In section 3.2, your arguments related to trap density effect on efficiency:
Can the author(s) explain how the increase defect density at higher bandgap can increase the efficiency? Are the defects (traps) deep or shallow and are they close to valence/conduction band.? Provide relevant reference to back your arguments.
3. In section 3.3, provide sufficie explanation (or contributing factors) to explain the shift (or higher) Voc for TiO2:N sample?
Author Response
I find the study relevant for the perovskite research community encompassing solar cells and LEDs, etc. I have a few comments on the results and discussion as noted below which need to be addressed.
- If possible, provide a schematic band-diagram and explain how the large NiO bandgap can enhance the PCE, considering the bandgap changes from 2.5 eV to 3 eV
Response: The authors would like to thank the reviewer for the valuable suggestion. However, we wish to notify that one of the authors (Dr Kuraganti Vasu) determined the bandgap of TiO2:N layer through experiments as shown below in one of his earlier analysis [50].
The previous research reports that an n-type anatase TiO2 with a bandgap of 3.23eV has experienced a drop in fermi energy level and bandgap to 3.07 eV (reported from Tauc plots) when doped with 4.0% nitrogen concentration (reported XPS analysis); also, becoming a p-type layer [50].
Hence, in the present simulation it is observed that power conversion efficiency increases as the HTL bandgap raise beyond 2.5 eV. However, for bandgap lower than 2.5 eV, simulation resulted a saturation pattern on the I-V curve.
Authors address the above explanation at appropriate section in the revised manuscript (Page:5, Lines: 196-208)
- In section 3.2, your arguments related to trap density effect on efficiency:
Can the author(s) explain how the increase defect density at higher bandgap can increase the efficiency? Are the defects (traps) deep or shallow and are they close to valence/conduction band.? Provide relevant reference to back your arguments.
Response: The authors would like to thank the reviewer for the valuable suggestion.
We wish to notify that the defect density is in accordance with the interface between the absorber layer and the ETL (significance explained in section 3.2) but not the absorber layer and the HTL. This concept is explained with reference to [64,65]. This simulation shows that, even if a defect density incurs in the interfacial layers, a HTL with higher bandgap can sustain the charge transport, avoid recombination and result in comparatively higher efficiencies.
- In section 3.3, provide sufficient explanation (or contributing factors) to explain the shift (or higher) Voc for TiO2:N sample?
Response: The authors would like to thank the reviewer for the valuable suggestion.
From the simulation results reported in section 3.3, it is observed that the value of Voc for TiO2:N based HTL PSC structure is similar to literature results observed for Spiro-OMeTAD and PEDOT:PSS structures. A shift in power conversion efficiency for proposed TiO2:N based HTL PSC structure is observed due to change in charge transport mobility and CB & VB energy density of states. The results of present study agree with the experimental data reported in literature as shown in Table 6.

Round 2
Reviewer 2 Report
The revision is unsatisfactory.
The authors have added two paragraphs. Paragraph 1 explains the meaning of the research and cites additional papers. Paragraph 2 describes material preparation, citing more additional papers. These only give context and add citations. There was no change in experimental device current data from the authors. The previous report demanded more data to support the authors’ claims on the following points.
1. Fig. 1 misuses energy band diagrams
The flat band energy bandgaps have not been converted to an energy band diagram.
2. Fig. 2 only shows illuminated cases, no dark current
No dark current data has been added to figures.
3. Conversion to device current missing, should be nontrivial
No calibration of device current from theory to experimental data has been performed.
4. Perovskite doping, direct/indirect missing
No perovskite semiconductor properties have been added to figures.
5. Missing discussion of device contacts
No device contact discussion has been added to figures.
The previous report demanded more justification for device current data in the form of more data from the authors. Instead, they have only cited other peoples’ work. It cannot be used to claim systematic calibration between theory and experiment. This paper must be rejected.
Author Response
Please
